# Mitigating Overestimation in Offline Reinforcement Learning with Anomaly Detection

## Abstract

Reinforcement Learning (RL) encounters substantial challenges in real-world applications, due to the time-consuming, costly, and risky nature of interacting with the environment. Offline Reinforcement Learning addresses this limitation by training models on static datasets, allowing an optimal policy to be learned from pre-collected data without requiring additional interactions with the environment. However, in this setting, when the agent queries actions outside the training data distribution, it can lead to overestimation of Q-values for OOD (Out-of-distribution) actions, ultimately hindering policy optimization. Previous works attempted to address this problem using explicit constraints such as penalty terms or support restriction. But these methods often fail to identify OOD actions or result in overly conservative Q-value estimates. We propose a novel solution that adjusts weights during training by using an anomaly detection model to identify the distribution of the offline dataset and employing anomaly scores to guide the offline RL process. Our method(RLAD) not only effectively mitigates the overestimation of OOD actions but also achieves near state-of-the-art performance on continuous D4RL tasks. Additionally, this framework is highly flexible, allowing for integration with various off-policy or offline RL algorithms and Anomaly Detection models to enhance performance.

## 1 Introduction

Reinforcement Learning (RL) has become a pivotal field in artificial intelligence, demonstrating significant achievements in areas such as natural language processing (NLP) (Ouyang et al., 2022) and computer vision ((Furuta et al., 2019), (Liu et al., 2023)). Its ability to enable agents to interact with environments and perform human-like sequential decision-making has made RL particularly valuable in fields such as robotics (Vecerik et al., 2018), medical diagnosis, and autonomous driving (Gu et al., 2023). However, applying reinforcement learning (RL) to real-world problems poses significant challenges due to the time-consuming, costly, and risky nature of continuous environment interactions.

To address these challenges, offline Reinforcement Learning (Offline RL) has emerged, focusing on leveraging static datasets for training. Unlike traditional RL, Offline RL lacks the continuous exploration that helps the agent converge to an optimal policy. Additionally, Offline RL must work with limited datasets, often failing to provide sufficient coverage of all possible state-action pairs above given offline dataset leading to a phenomenon known as distributional shift. This limitation leads to a prominent issue: the overestimation of the Q-function, especially for state-action pairs not present in the training dataset, as highlighted by the work of (Kumar et al., 2019). Methods like CQL, BCQ, and BRAC ((Kumar et al., 2020), (Fujimoto et al., 2019), (Wu et al., 2019) have been developed to tackle this problem by making value functions pessimistic or restrict the action space of a policy, which can result in overly pessimistic evaluations or overly restrictive regions. Therefore, many works try to make the agent not too conservative or to address distributional shift and OOD actions while maintaining trading off with the previous distributional shift and OOD repression to obtain an optimal policy. ((Lyu et al., 2022), (Hong et al., 2023))

Inspired by the analysis of (Kumar et al., 2019), which highlights that overestimation mainly occurs with data outside the training distribution, and the fact that explicit constraints can result in over-pessimistic value estimation, we have focused on the following point: How can we better utilize the distribution of the training dataset to avoid overestimation caused by OOD data? While we aim to prevent the agent from querying out-of-distribution actions, we also recognize that not every out-of-distribution action leads to a sub-optimal policy. Therefore, we seek to avoid imposing hard or explicit constraints.

We propose that anomaly detection models, which are highly effective modules for identifying out-of-distribution or novel data, can be a possible novel solution. These models adept at classifying binary problems based on anomaly scores, which typically quantify how much a data point deviates from the training distribution. These scores can provide useful guidance for the agent during training, so we can plug-in any off-policy algorithm without the need to introduce additional regularizers into their objective function.

In this context, we propose approaching the Offline RL problem through the lens of a Anomaly Detection, classifying data as either in-distribution (normal) or out-of-distribution (abnormal). Our proposed method, **Reinforcement Learning with Anomaly Detection (RLAD)**, addresses these challenges by leveraging deep learning-based anomaly detection models ((Zong et al., 2018), (Ruff et al., 2018)) to efficiently and accurately estimate whether unseen data belongs to the in-distribution or out-of-distribution. This approach capitalizes on the expressive power of neural network networks without the need for specialized regularizer to match distributions, support or underestimate Q-value. Our algorithm operates in two stages:

1) **Anomaly Detection Model Training**: Train the anomaly detection model on the training dataset. During this phase, the model learns which data points are close to the training distribution and assigns an anomaly score as an indicator.

2) **Anomaly Score-Based Weight Adjustment**: Use the anomaly score of the given data to inform the agent, adjusting the importance of each sample during learning through the application of different sample weights.

Our experiments aim to demonstrate about anomaly scores(weight) and estimated Q-values. Our algorithm performs well in continuous environments, achieving state-of-the-art performance in several tasks. Our contributions are as follows:

1) **Framework**: We present a simple, easily implementable framework that can be integrated with various off-policy or offline RL algorithms and anomaly detection models, especially according to the environment.

2) **Q-value Estimation**: We demonstrate that Anomaly Score-Based Weight Adjustment significantly contribute to accurate Q-value estimation and the mitigation of overestimation.

3) **Empirical Performance**: We show good empirical performance, particularly in MuJoCo environments, achieving state-of-the-art results.

## 2 RELATED WORKS

**Policy Constraints** Several works have proposed imposing constraints on actor-critic algorithms by constraining the learned policy to remain close to the behavior policy that generated the offline dataset $D$. (Fujimoto et al., 2019) directly estimates the behavior policy $\pi_\beta$ and constrains the learned policy $\pi_\theta$ to stay close to the estimated $\pi_\beta$. In contrast, (Kumar et al., 2019) argues that support matching using Maximum Mean Discrepancy (MMD) is more effective than direct matching of distributions.

Additionally, (Wu et al., 2019) brings $\pi_\beta$ closer to the target policy by penalizing the critic, leading to more pessimistic evaluations. Methods like (Peng et al., 2019) and (Nair et al., 2021) use the advantage term to construct an implicitly weighted maximum likelihood objective, while others, such as (Fujimoto & Gu, 2021), operate without the actor-critic structure. These approaches aim to fundamentally prevent OOD actions from being selected by the learned policy $\pi_\theta$, thus mitigating overestimation and providing relatively stable performance. However, these strategies can sometimes excessively constrain or regularize actions of the agent.

**Conservative Value Estimation** Rather than restricting the policy, some methods focus on conservatively estimating the value function of state-action pairs by explicitly penalizing unseen OOD actions in $D$ (Kumar et al., 2020) or by avoiding directly querying unseen actions in the dataset using expectile regression to maximize the advantage (Kostrikov et al., 2021). Additionally, model-based approaches such as (Yu et al., 2022) learn dynamic models to interpolate and augment data, combined with conservative Q-value estimation. While these works aim to avoid the OOD overestimation problem by penalizing or ignoring unseen OOD actions in $D$, they often suffer from overly pessimistic value estimates.

Other approaches, such as (Gal & Ghahramani, 2016), (Wu et al., 2021), (An et al., 2021), (Bai et al., 2022), utilize uncertainty measures—like Bayesian methods, MC-dropout, and ensemble to adaptively adjust the level of OOD suppression. These strategies can operate without explicit policy constraints, but the computational cost of measuring uncertainty can be high and incorrect.

**Trajectory Optimization** Recently, Conditional Sequence Modeling (CSM) has emerged as a novel paradigm for RL tasks, associating individual trajectories with return-to-go (RTG) tokens, enabling the handling of long sequences with large model sizes, such as in (Janner et al., 2021) and (Chen et al., 2021). However, due to the intrinsic stochasticity of state transitions, approaches like (Yamagata et al., 2023) and (Chebotar et al., 2023) have been proposed to address these limitations, particularly in offline settings.

**Anomaly Detection** Anomaly Detection (AD) is the task of identifying samples that deviate significantly from the majority of the data, often signaling irregular, fake, rare, or fraudulent observations (Wang et al., 2019). Particularly, semi-supervised AD is defined as the task of detecting samples that are out of distribution by using only normal samples in the training dataset. To estimate the normal distribution and detect exceptional samples from that distribution, a wide range of models is available, spanning from statistical to deep-learning based AutoEncoder ((Ruff et al., 2018), (Zong et al., 2018)) Transformer (Xu, 2021), and flow-based(Zhou et al., 2024). These methods have been improved to handle various problem of each data types, including images, video, and time-series data.

Given the dimension reduction capabilities and generalizability of AutoEncoders in handling the varying space sizes of reinforcement learning datasets, we select two models for the AD module to capture distributional information: Deep SVDD ((Ruff et al., 2018)) and DAGMM ((Zong et al., 2018)).

**Deep SVDD** Deep SVDD (Ruff et al., 2018) is a deep learning-based extension of the classical Support Vector Data Description method.(Tax & Duin, 2004) Unlike SVDD, which uses hand-crafted kernels such as the Gaussian kernel, Deep SVDD learns the appropriate feature space through a deep neural network with an AutoEncoder architecture. The goal of Deep SVDD is to find the smallest hypersphere that encloses most of the normal data in the feature space by mapping inputs through the model.

**DAGMM** Similar to Deep SVDD, DAGMM also utilizes an AutoEncoder to encode data into a feature space.(Zong et al., 2018) However, unlike Deep SVDD, which seeks to find the optimal hypersphere for normal data, DAGMM estimates the distribution of data in the latent space by utilizing a Gaussian Mixture Model.

## 3 PRELIMINARIES

### 3.1 REINFORCEMENT LEARNING

We formulate Reinforcement Learning using the standard Markov Decision Process (MDP), $(S, A, P, R, \gamma)$, where $S$ represents the state space, $A$ denotes the action space, $P(s'|s, a)$ is the transition probability, $R(s, a) : S \times A \to \mathbb{R}$ is the reward function, and $\gamma \in (0, 1]$ is the discount factor. Reinforcement Learning aims to find an optimal policy that maximizes the expected cumulative discounted reward, $\mathbb{E}_\tau[\sum_{t=0}^{\infty} \gamma^t R(s_t, a_t)]$, where $\tau$ is a trajectory.

There are several approaches to finding an optimal policy, such as Policy Gradient (Sutton et al., 1999), Q-learning (Sutton & Barto, 2018), and Actor-Critic methods (Konda & Tsitsiklis, 1999). Among them, Q-learning and Actor-Critic methods are based on the Bellman Equation:

$$Q^\pi(s, a) = R(s, a) + \gamma \mathbb{E}_{P(s'|s,a)}[V^\pi(s')]$$

where $Q^\pi(s, a) = \mathbb{E}_\pi[\sum_{t=0}^\infty \gamma^t R(s_t, a_t)|s = s_t, a = a_t]$ is the state-action value function, and $V^\pi(s) = \mathbb{E}_\pi[\sum_{t=0}^\infty \gamma^t R(s_t, a_t)|s = s_t]$ is the state value function. Both methods update the Q-function via the Bellman operator $\mathcal{B}$:

$$(\mathcal{B}Q^\pi)(s, a) = R(s, a) + \gamma \mathbb{E}_{P(s'|s,a)}[\mathbb{E}_\pi[Q^\pi(s', a')]]$$

In this paper, we utilize the Actor-Critic method, a widely used approach that alternates between policy evaluation and policy improvement, which demonstrates faster and more efficient convergence towards finding the optimal policy.

## 3.2 Offline Reinforcement Learning

In real-world applications, interacting with the environment can often be risky, expensive, and time-consuming. To address these challenges, Offline Reinforcement Learning (Offline RL) has been developed, which leverages a pre-collected dataset to learn an optimal policy without additional environment interaction. The pre-collected dataset consists of multiple trajectories $\tau = (s_0, a_0, r_0, s_1, a_1, r_1, ...)$.

However, the lack of interaction with the environment makes it difficult for the agent to explore new state-action pairs, leading to the domain shift problem, which results in a sub-optimal policy. Additionally, standard off-policy algorithms often perform poorly in the offline setting, leading to a need for new methods. Many approaches focus on mitigating the overestimation problem caused by out-of-distribution actions through value function or policy constraints, uncertainty-based methods, or model-based approaches. In this paper, we aim to address this issue using anomaly scores from independently trained anomaly detection module.

## 4 Reinforcement Learning with Anomaly Detection (RLAD)

In this paper, we propose addressing the Offline RL problem through the lens of Anomaly Detection, which we refer to as Reinforcement Learning with Anomaly Detection (RLAD). Offline Reinforcement Learning encounters the overestimation problem primarily when the agent bootstraps actions that are not present in the training dataset. As a result, the accumulation of bootstrapping errors can significantly degrade performance.

To address the issue of out-of-distribution actions being bootstrapped, we provide an anomaly score as guidance to the agent to indicate how far a data point is from the distribution of the training dataset. Specifically, anomaly scores are used as weights for the objective functions of both the Actor and Critic. Our approach has empirically shown promising results.

## 4.1 Why Anomaly Detection?

As mentioned earlier, previous works in offline reinforcement learning algorithms often solve the overestimation problem by imposing constraints on value functions or policies. However, these methods can lead to overly conservative value estimates or overly restrictive action spaces, resulting in sub-optimal policies. Moreover, approaches based on ensemble or dropout-based uncertainty measures, being suggested to solve this problem, have inevitable problem of high computational cost or inaccurate estimation of uncertainty.

In contrast, Anomaly Detection models are effective at detecting out-of-distribution data with respect to the normal training dataset. These models typically determine whether data points are in-distribution or out-of-distribution based on anomaly scores.

In this reason, we propose that anomaly scores can serve as efficient and effective guidance for the agent to determine whether a given action is out-of-distribution, helping the agent adjust its policy accordingly.

## 4.2 Pretraining the Anomaly Detection Model

To inform the agent of how far the data is from the training set, we first train an anomaly detection model. In this paper, we use Deep SVDD and DAGMM with state-action pairs as the input. Although

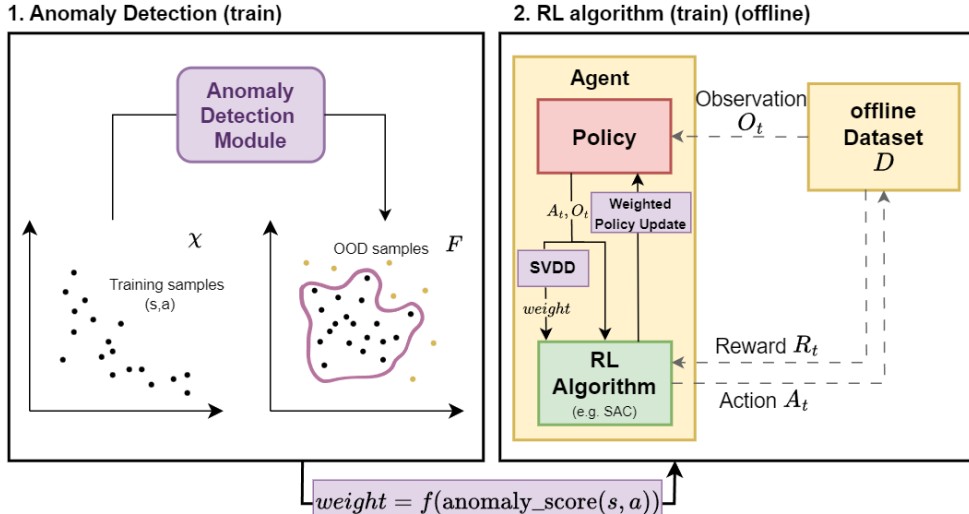

Figure 1: This figure illustrates the overall pipeline. **Anomaly Detection Train(left)** First train the Anomaly Detection(AD) module using offline Reinforcement Learning dataset $D$. **RL Train(right)** Then weighted update is performed for RL algorithm update for every iteration with anomaly score from AD module and scaling function $f$. For offline update, every sample is bootstrapped from offline Dataset $D$.

we considered using full trajectories, capturing the features of each trajectory proved challenging, so we opted for state-action pairs to simplify assumptions and efficiently capture features. We denote the anomaly score for a state-action pair $(s, a)$ as $anomaly\_score(s, a)$. When the state $s$ is fixed, this can be roughly interpreted with behavior policy. Additionally, we assume that the further a state-action pair is from the dataset, the higher the $anomaly\_score(s, a)$.

### 4.3 TRAINING THE RL AGENT

Using the previously trained Anomaly Detection model, we compute the anomaly score for each state-action pair. This score is then used to weight the value function training and policy optimization. Since we want the agent to focus more on data points that are closer to the training distribution, the weight should increase as the anomaly score decreases. A higher weight indicates a lower likelihood of being OOD, encouraging more learning. Conversely, a lower weight indicates a higher likelihood of being OOD, reducing its impact on learning.

This approach avoids imposing strict penalties or limiting the agent to the training distribution, effectively mitigating the overestimation problem caused by OOD data while still allowing for accurate estimation of OOD actions. The overall algorithm can be summarized as follows:

1) **Calculate Anomaly Score**: Use the trained Anomaly Detection model to calculate the anomaly score for the next state-action pair. Since the agent only queries the next action, we need to train the network based on the next state-action pair.

2) **Compute Weight**: Determine the weight from pretrained Anomaly Detection model as follows:
$$weight(s, a) := f(anomaly\_score(s, a))$$
where $f(\cdot)$ is a non-negative, monotonically decreasing function bounded within the range of $anomaly\_score(\cdot, \cdot)$. The specific form of $f(\cdot)$ may vary depending on the model used to compute the anomaly score. In our implementation, we use $f(x) = \frac{1}{x}$ for Deep SVDD and $f(x) = sigmoid(-x)$ for DAGMM.

3) **Weighted Q-Function Training**: Train the Q-function using the computed weights with the following objective function:
$$\nabla_\theta \mathcal{L}(\theta) = \mathbb{E}[weight(s', a')(Q_\theta(s, a) - r - Q^*(s', a'))\nabla_\theta Q_\theta(s, a)]$$

where $Q^*$ is the target value function.

4) **Weighted Policy Training**: Train the policy using the computed weights with the following objective function:

$$\nabla_\phi \mathcal{L}(\phi) = \mathbb{E}[weight(s,a)(Q_\theta(s,a)\nabla_\phi \log \pi_\phi(a|s))]$$

We provide the pseudo-code for this algorithm in Algorithm 1.

---

**Algorithm 1** RLAD Training Procedure (Actor-Critic Style)

---

**Input:** Pre-collected dataset $D = \{(s,a,r,s')\}$, `Anomaly Detection` model $\psi$, SAC model with Q-function $Q_\theta$ and policy $\pi_\phi$

hyperparameters: learning rates $\alpha_\phi, \alpha_\theta$, discount factor $\gamma$, delayed update rate $\tau$

**Output:** Trained SAC model

1: **1. Pretrain the Anomaly Detection model $\psi$ on the dataset $D$**
2: **2. Training the Actor-Critic model with weights from Anomaly Detection model**
3: Initialize value function parameters $\theta$ and policy parameters $\phi$
4: Initialize learning rates $\alpha_\theta$ and $\alpha_\phi$
5: Initialize target parameters equal to Q-function parameters $\theta' \leftarrow \theta$ and policy parameters $\phi' \leftarrow \phi$
6: $weight(s,a) := f(anomaly\_score(s,a))$     ▷ $f$ is a bounded, non-negative function which is monotonic decreasing
7: **while** not converged **do**
8:     **for** each $(s,a,r,s')$ in $D$ **do**
9:         Compute target Q-value:

$$y = r + \gamma Q_{\theta'}(s', a' \sim \pi_{\phi'}(\cdot|s'))$$

10:         Compute weighted Q-function loss:

$$\nabla_\theta \mathcal{L}(\theta) = \mathbb{E}[weight(s',a')(Q_\theta(s,a) - r - Q_{\theta'}(s',a'))\nabla_\theta Q_\theta(s,a)]$$

11:         Update Q-function parameters:

$$\phi \leftarrow \phi - \alpha_\phi \nabla_\phi \mathcal{L}_Q$$

12:         Compute weighted policy loss:

$$\nabla_\phi \mathcal{L}(\phi) = \mathbb{E}[weight(s,a)(Q_\theta(s,a)\nabla_\phi \log \pi_\phi(a|s))]$$

13:         Update policy parameters:

$$\theta \leftarrow \theta - \alpha_\theta \nabla_\theta \mathcal{L}_\pi$$

14:     **end for**
15:     Update target networks periodically:

$$\phi' \leftarrow \tau\phi + (1-\tau)\phi'$$
$$\theta' \leftarrow \tau\theta + (1-\tau)\theta'$$

16: **end while**
17: **return** Trained Actor-Critic model

---

# 5 EXPERIMENTAL RESULTS

In this section, our experiments try to address the following questions:

1. Does our framework achieve better performance with various combinations of Anomaly Detection models and Reinforcement Learning algorithms?

2. Does our framework estimate Q values correctly in the training distribution and do not estimate Q values too conservative in the out-of-distribution?

To answer the above questions we present two experiments : First, we analyze Q value errors (between the true value and the estimated value from offline algorithms) with weights on the training distribution and the out-of-distribution. Second, we compare normalized returns with some baselines in some tasks MuJoCo environment including medium, medium-replay and medium-expert and Adroit environment including human and cloned. Also we compare Q-value distributions among Soft Actor-Critic, RLAD-SAC-SVDD, and CQL in 'Pendulum-v1' environment.

## 5.1 ANALYSIS OF WEIGHT AND Q DIFFERENCE

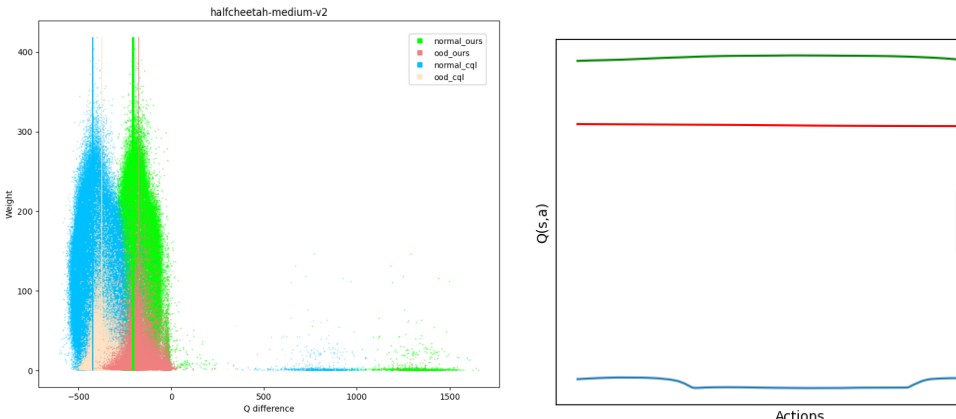

Figure 2: Scatter plot for weight-$Q_{\text{difference}}$    Figure 3: Q values of the entire action space in 'Pendulum-v1'

In this section, we analyze the relationship between the weight and the $Q_{\text{difference}}$ to demonstrate the effectiveness of our method.

To estimate $Q^*(s, a)$, we used the critic network of an SAC model trained in online setting for proxy. We then compared the $Q(s, a)$ values from both our model and the baseline model, CQL, with simple qunaitifing measure defined as $Q_{\text{difference}}(s, a) := Q(s, a) - Q^*(s, a)$. Additionally, we extracted an OOD dataset using AutoEncoder and MC-Dropout, independent of the offline dataset and Anomaly Detection module, to analyze OOD actions and Q-value differences. For the validity of these datasets, please refer to t-SNE visualizations in Appendix, figure 5.

The figure shows the distribution of weight as a function of $Q_{\text{difference}}$ for both normal and OOD samples across our model(RLAD-SAC) and CQL in the 'halfcheetah-medium-v2' environment. Notably, both models display a peak at negative $Q_{\text{difference}}$ values, with a long tail extending toward positive values. However, our model exhibits a more concentrated peak near $Q_{\text{difference}} = 0$, while CQL shows more spread in the positive direction, indicating more frequent overestimations.

For OOD samples, the distribution of our model shows that $Q_{\text{difference}}$ are near zero and weights from the anomaly detection module are smaller than normal data, which demonstrates that the anomaly detection module performs well. In contrast, $Q_{\text{difference}}$ of CQL are far from zero for OOD samples, suggesting overly conservative estimation.

Table 1: Comparison of normalized average returns of RLAD against original SAC model (offline) and Behavior Cloning(BC) on the D4RL MuJoCo Gym datasets. Normalized returns of RLAD are averaged over 5 random seeds, and baseline values are taken from each paper.

| Task Name | RLAD-SAC-SVDD (OURS) | RLAD-SAC-DAGMM (OURS) | SAC | BC |
|---|---|---|---|---|
| hopper-med | **105.48**±3.24 | 63.68±5.58 | 0.8 | 29.0 |
| walker2d-med | 95.18±5.22 | **114.92**±7.36 | -0.3 | 75.3 |
| halfcheetah-med | 75.95±8.63 | 77.83±5.58 | 55.2 | 36.1 |
| hopper-med-rep | **106.10**±2.00 | 92.74±14.92 | 7.4 | 11.8 |
| walker2d-med-rep | **105.00**±6.10 | 99.74±12.21 | -0.3 | 11.3 |
| halfcheetah-med-rep | **79.43**±4.81 | 71.78±4.55 | 0.8 | 38.4 |
| hopper-med-exp | **105.01**±4.59 | 98.70±7.03 | 0.7 | 53.9 |
| walker2d-med-exp | 104.92±8.56 | **105.32**±10.39 | 1.9 | 36.9 |
| halfcheetah-med-exp | 78.05±2.39 | 73.39±7.65 | 28.4 | 35.8 |

In comparing the two methods, our model shows a more accurate Q-value estimation, particularly around $Q_{\text{difference}} = 0$, where weight peaks closer to this value indicate higher accuracy in policy learning. CQL, however, suffers from both under- and overestimation, as evidenced by its broader distribution and more pronounced presence of large positive $Q_{\text{difference}}$ values, even for normal data.

Also, as shown in table 1, this indicates that AD modules like Deep SVDD and DAGMM can capture the characteristics of the data and successfully estimate Q-values for samples not present in $D$, without directly imposing constraints on policy or value estimation. Moreover, the superior performance of RLAD-SAC (with Deep SVDD and DAGMM) over standard SAC highlights the potential for exploring a broader range of AD models.

## 5.2 BRIEF ANALYSIS ON Q-FUNCTIONS

We analyze the Q-functions of Online Soft Actor-Critic, RLAD-SAC-SVDD (Offline), and CQL (Offline) in the Pendulum-v1 environment, where the offline dataset is obtained by a random policy. The Pendulum environment is chosen for visualization on a 1D plane, eliminating the need for dimensionality reduction methods for clearer interpretation.

As shown in Figure 3, both our model and CQL estimate the Q-values conservatively. However, CQL exhibits more conservative estimates than our model, while our model provides more accurate Q-value estimations.

## 5.3 COMPARISON WITH D4RL

Based on the above analysis showing the effectiveness of our model, we compare our method to prior offline RL methods on continuous domains and dataset compositions. We implement our method based on Soft Actor-Critic(Haarnoja et al., 2018) and BEAR(Kumar et al., 2019) for Reinforcement Learning algorithms and Deep-SVDD(Ruff et al., 2018) and DAGMM (Zong et al., 2018) for Anomaly Detection model.

**Evaluation on D4RL** Results for the MuJoCo tasks and Adroit tasks (Rajeswaran et al., 2018) in the D4RL benchmarks (Fu et al., 2021) are shown in 2 and 3, respectively. The results for other baselines are based on their respective papers ((Kang et al., 2023), (Kidambi et al., 2020), (Rigter et al., 2022), (Kumar et al., 2019), (Cheng et al., 2022), (Kumar et al., 2020), (Kostrikov et al., 2021), (Chen et al., 2021)). We evaluate three environments of MuJoCo tasks, hopper, walker2d, and halfcheetah, with three dataset types, medium, medium-replay, and medium-expert, for each environment. Also, we evaluate four environments of Adroit tasks, pen, hammer, door, and relocate, with two data types, human and cloned. We achieve the best performance on most environments. Especially, we obtain good performance on the multimodal datasets such as medium-expert or medium-replay. This implies that our method is robust to the complex distribution.

Table 2: Comparison of normalized average returns of RLAD against baselines on the D4RL MuJoCo Gym datasets. Normalized returns of RLAD are averaged over 5 random seeds, and baseline values are taken from each paper.

| Task Name | RLAD-SAC-SVDD (OURS) | RLAD-BEAR-SVDD (OURS) | RLAD-SAC-DAGMM (OURS) | EDP (TD3+BC) | MOReL | RAMBO | BEAR | ATAC | CQL | IQL | DT |
|---|---|---|---|---|---|---|---|---|---|---|---|
| hopper-med | **105.48**±3.24 | 95.95±4.90 | 63.68±5.58 | 81.9 | 95.4 | 92.8 | 30.77 | 85.6 | 86.6 | 66.3 | 67.6 |
| walker2d-med | 95.18±5.22 | 90.33±8.69 | **114.92**±7.36 | 86.9 | 77.8 | 86.9 | 56.02 | 89.6 | 74.5 | 78.3 | 74.0 |
| halfcheetah-med | 75.95±8.63 | 45.10±0.94 | **77.83**±5.58 | 52.1 | 42.1 | 77.6 | 37.14 | 53.3 | 44.4 | 47.4 | 42.6 |
| hopper-med-rep | **106.10**±2.00 | 89.35±10.58 | 92.74±14.92 | 101.0 | 93.6 | 96.6 | 31.13 | 102.5 | 48.6 | 94.7 | 82.7 |
| walker2d-med-rep | **105.00**±6.10 | 66.61±6.45 | **99.74**±12.21 | 94.9 | 49.8 | 85.0 | 13.66 | 92.5 | 32.6 | 73.9 | 66.6 |
| halfcheetah-med-rep | **79.43**±4.81 | 42.16±1.15 | 71.78±4.55 | 49.4 | 40.2 | 68.9 | 36.21 | 48.0 | 46.2 | 44.2 | 36.6 |
| hopper-med-exp | 105.01±4.59 | **113.34**±1.55 | 98.70±7.03 | 97.4 | 108.7 | 83.3 | 67.26 | 111.9 | 111.0 | 91.5 | 107.6 |
| walker2d-med-exp | 104.92±8.56 | 96.34±4.62 | **105.32**±10.39 | 110.2 | 95.6 | 68.3 | 43.80 | **114.2** | 98.7 | 109.6 | 108.1 |
| halfcheetah-med-exp | 78.05±2.39 | 92.90±0.62 | 73.39±7.65 | **95.5** | 53.3 | 93.7 | 44.16 | 94.8 | 62.4 | 86.7 | 86.8 |

Table 3: Comparison of normalized average returns of RLOCC against baselines on the D4RL Adroit datasets. Normalized returns of RLAD are averaged over 5 random seeds, and baseline values are taken from each paper.

| Task Name | RLAD-SAC-SVDD (OURS) | ATAC | CQL | ARMOR | IQL | BC |
|---|---|---|---|---|---|---|
| pen-human | **84.09**±9.63 | 53.1 | 37.5 | 72.8 | 71.5 | 34.4 |
| hammer-human | **9.29**±6.21 | 1.5 | 4.4 | 1.9 | 1.4 | 1.5 |
| door-human | **16.18**±5.42 | 2.5 | 9.9 | 6.3 | 4.3 | 0.5 |
| relocate-human | 0.04±0.04 | 0.1 | 0.2 | 0.4 | 0.1 | 0.0 |
| pen-cloned | **56.17**±14.66 | 43.7 | 39.2 | 51.4 | 37.3 | **56.9** |
| hammer-cloned | **2.75**±2.83 | 1.1 | 2.1 | 0.7 | 2.1 | 0.8 |
| door-cloned | 0.72±0.98 | 3.7 | 0.4 | -0.1 | 1.6 | -0.1 |
| relocate-cloned | 0.08±0.29 | 0.2 | -0.1 | -0.0 | -0.2 | -0.1 |

## 5.4 CONCLUSION

By analyzing Q differences, we demonstrate that our model which utilizes Anomaly Detection(AD) models, effectively mitigates the overestimation of OOD samples compared to prior offline RL algorithms, also do not estimate Q-values overly conservative. Moreover, in the D4RL benchmarks, our model achieves near state-of-the-art performance across various environments, for several combinations of Reinforcement Learning algorithms and Anomaly Detection models. These experiments highlight the potential of integrating Anomaly Detection modules to detect OOD samples and leveraging their quantitative anomaly scores as guidance to enhance performance. Because of the independence of AD module, this allows for the flexible integration of various models without imposing explicit constraints on policy or value functions, facilitating the exploration of optimal combinations between various branches of Anomaly Detection and powerful RL algorithms.

For future work, we aim to find the way how to choose the best combination of an Anomaly Detection module and a Reinforcement Learning algorithm according to the characteristic of the dataset or the environment. This research will enable more effective selection of an anomaly detection model and a reinforcement learning algorithm. We believe that our work provides a deeper empirical understanding of the relationship between accurate Q-value estimation and overall performance across various environments and RL algorithms. Additionally, we see potential for our approach to be further developed, much like data type-specific AD research in areas such as images and time series.

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

## A APPENDIX

### A.1 IMPLEMENTATION DETAILS

This is hyper-parameters for SAC and BEAR.

Table 4: Hyperparameters

| Hyperparameter | Value/Type |
|---|---|
| model hidden size | 256 |
| number of layers | 2 |
| policy learning rate | 1e-4 |
| critic learning rate | 3e-4 |
| epochs | 3000 |
| optimizer | Adam |
| batch size | 256 |
| target network update rate ($\tau$) | 1e-2 |
| discount factor | 0.99 |
| sampled actions for MMD (bear) | 100 |
| number of q functions (bear) | 2 |
| kernel type (bear) | gaussian kernel |

Our code is based on '`https://github.com/rail-berkeley/rlkit.git`'.

### A.2 OFFLINE RL ALGORITHMS

We implement our method on top of Soft Actor-Critic(SAC)(Haarnoja et al., 2018) and Bootstrapping Error Accumulation Reduction(BEAR)(Kumar et al., 2019). Since both them are actor-critic methods, we apply Anomaly Score-based sample weights to both the actor and critic loss functions.

The following is hyperparameters for Deep SVDD and DAGMM:

Table 5: Hyperparameters for Deep SVDD

| Hyperparameter | Value/Type |
|---|---|
| hidden dimension | 256 |
| latent dimension | 128 |
| optimizer | Adam |
| learning rate | 1e-3 |
| batch size | 256 |
| epochs | 500 |
| weight decay ($l^2$) | 0.5e-6 |

Table 6: Hyperparameters for DAGMM

| Hyperparameter | Value/Type |
|---|---|
| hidden dimension | 256 |
| latent dimension | 2 |
| optimizer | Adam |
| learning rate | 5e-4 |
| batch size | 256 |
| epochs | 200 |
| # of gaussian distributions | 4 |
| $\lambda_{enery}$ | 0.1 |
| $\lambda_{cov\_diag}$ | 0.005 |

## A.3 EXPERIMENT DESIGN : ANALYSIS OF WEIGHT AND Q DIFFERENCE

In 5.1, our experiment procedure is as the following: we train the Anomaly Detection model and train two agents based on SAC which are in online setting and offline setting, respectively.

1. **Offline Learning and Proxy for Optimal Q-Value: Online Learning** To estimate $Q^*(s, a)$, we utilized the critic network of a Soft Actor-Critic (SAC) model trained with a large amount of online data as a proxy for the optimal policy and the optimal policy evaluation network (critic).

2. **Sampling and calculating Q difference** Afterward, we trained both our proposed model and the baseline model, CQL, in an offline setting. We then compared the difference between the $Q(s, a)$ values obtained from the critic networks of both offline models with the $Q^*(s, a)$ values from the online SAC critic, visualizing these differences in a graph(2).

$$Q_{\text{difference}} = (f(anomaly\_score(s, a)), Q^*(s, a) - Q(s, a))$$

3. **Sampling** $D_{ood}$ Furthermore, to analyze the anomaly scores of OOD actions and the differences in $Q$-values, we extracted an OOD dataset,$D_{ood}$, using Autoencoder(AE) and Monte Carlo Dropout(MC-Dropout). For ensuring the objectivity of test, the extraction of $D_{ood}$ is performed independently of both the offline dataset $D$ and the Anomaly methods. Detailed procedure is:

   - Training the autoencoder model with dropout layers using the normal set $D$.
   - For each data in the training dataset, calculate the variance of the reconstruction error with Monte Carlo(MC) Dropout. Set the threshold as 10 times of the maximum of the variance of the reconstruction error within the training dataset.
   - With the random policy, calculate the variance of the reconstruction error for each state, action pairs in an episode with MC dropout. If the variance exceeds the threshold, we classify that state-action pair as out-of-distribution data and include it in the out-of-distribution dataset.

   Each $D$ and $D_{ood}$ is visualized by TSNE(Van der Maaten & Hinton, 2008) for the high and varying dimensionality of each environment(5)

As you can see in the figure 4, both CQL and our model estimate Q-values accurately in the training dataset. Meanwhile, in the out-of-distribution, our model estimates Q-values more accurate than CQL, that is Q values of the most of data in the out-of distribution are close to 0. This result shows that our model effectively mitigates Q values and estimates accurately.

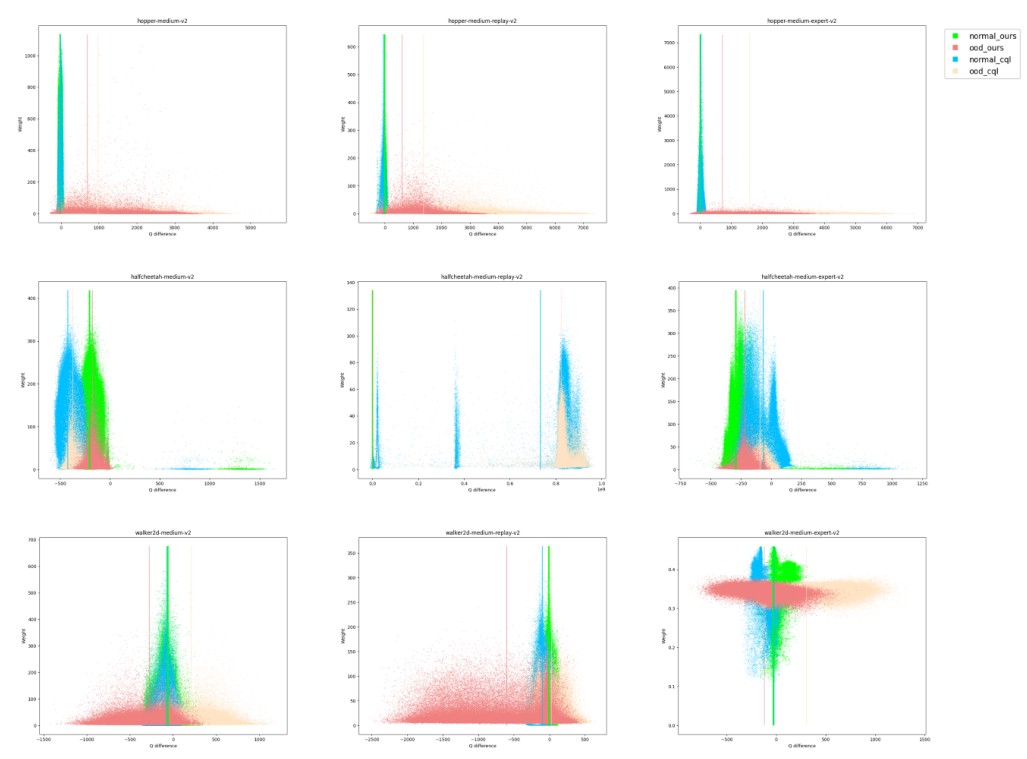

Figure 4: $Q_{\text{difference}}$ and Weight for each D4RL mujoco environments.

To describe the results of other mujoco environment, please see this figure 4. In the Hopper environment, the overall pattern is similar to the main figure, with the only difference being the gap between normal-CQL and normal-RLAD. For the halfcheetah-medium-replay environment, the results demonstrate that CQL's estimations for both the normal and OOD sets deviate significantly from the online proxy $Q^*$, whereas our model (green and pink) overlaps near the zero region, indicating more accurate estimations.

An exception to this pattern is observed in the halfcheetah-medium-expert environment, where the performance gap between RLAD-SAC and CQL is smaller than in other datasets. This could suggest that the dynamics of the expert dataset reduce the model's reliance on anomaly detection, partially explaining why RLAD-SAC's advantage is less pronounced in this setting. However, even in this case, RLAD-SAC avoids significant overestimation, particularly for the OOD set, which still provides a performance edge over CQL(2). Lastly, in the walker2d environment, both models perform well on the training dataset $D$, with our model showing a tendency toward underestimation, while CQL exhibits overestimation for $D_{ood}$ samples. This may further explain the performance gap between our model and CQL.

Interestingly, in the walker2d-medium-expert environment, the overall weight distribution is higher than in other environments. However, the AD module appears to be functioning well, as the OOD set still exhibits a lower peak in weight values compared to the training set. In this environment, not only our model could avoid overstimation of $D_{ood}$, but also estimate Q values for normal set $D$ more exactly.

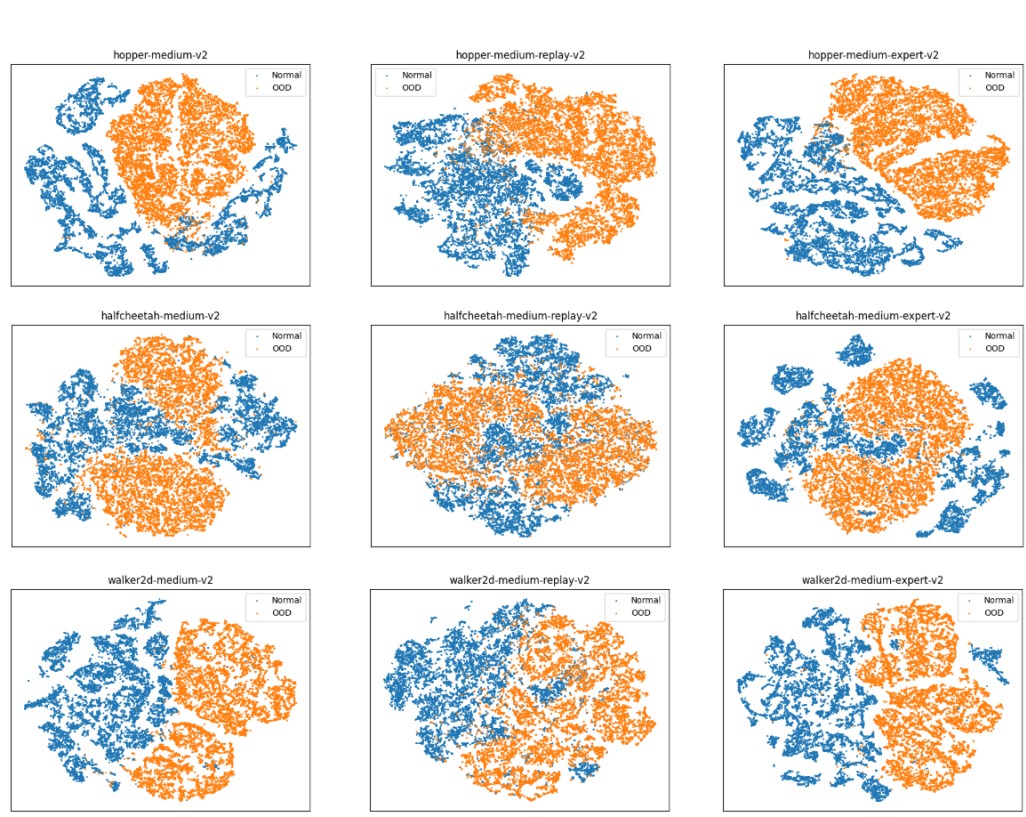

Figure 5: Distribution of offline dataset $D$(normal) and sampled $D_{ood}$(OOD).

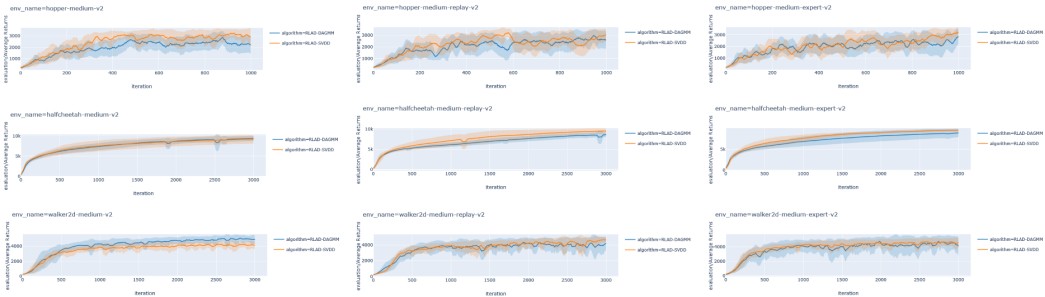

Figure 6: Learning curve of Deep SVDD guided SAC algorithm

## A.4 HARDWARES

For the Mujoco experiments, we use the medium, medium-replay, and medium-expert datasets for the hopper, walker2d, and halfcheetah environments, respectively. And for Adroit experiments, we use human and cloned for pen, hammer, door and relocate. To operate experiments, we use the following hardwares:

Table 7: Specifications of the Hardware Used for Experiments

| Attribute | Details |
|---|---|
| **CPU: Intel Xeon Silver 4210R** | |
| Model Name | Intel Xeon Silver 4210R @ 2.40GHz |
| Cores | 10 |
| Threads | 20 |
| Cache Size | 13.75 MB |
| Base Clock Speed | 2.40 GHz |
| Max Turbo Boost Speed | 3.20 GHz |
| **GPU: NVIDIA RTX 3090** | |
| Model | NVIDIA GeForce RTX 3090 |
| Number of GPUs | 4 |
| GPU Memory | 24 GB (each) |

