# OpenReview forum: "Mitigating Overestimation in Offline Reinforcement Learning with Anomaly Detection"
_ICLR.cc/2025/Conference — ICLR 2025 Conference Withdrawn Submission_

### Official Review · Reviewer_zp1L · 2024-10-30

**Soundness:** 2
**Presentation:** 1
**Contribution:** 2
**Rating:** 3
**Confidence:** 5

**Summary:**

The paper introduces Reinforcement Learning with Anomaly Detection (RLAD), an approach designed to address overestimation issues in offline reinforcement learning (RL). In offline RL, overestimation often arises when the agent evaluates actions outside the training distribution (out-of-distribution, or OOD), potentially leading to suboptimal policies. Traditional methods use explicit constraints or penalties to mitigate this but can be overly conservative. RLAD innovatively incorporates anomaly detection models to identify OOD actions and adjust weights during training based on anomaly scores, effectively guiding the RL model away from risky actions while preserving policy flexibility. RLAD employs deep learning-based anomaly detection models, such as Deep SVDD and DAGMM, to compute anomaly scores that serve as weights for training updates, reducing overestimation on OOD actions. Using these anomaly scores as weights, RLAD enhances Q-value accuracy by diminishing the impact of OOD actions on policy and value function updates, reducing the likelihood of overestimation. RLAD demonstrates good results in continuous control environments, including the MuJoCo suite.

**Strengths:**

* The use of anomaly detection models (e.g., Deep SVDD, DAGMM) to address overestimation in offline RL is innovative.
* Overestimation in offline RL can significantly hinder policy performance. By dynamically adjusting Q-value updates based on anomaly scores, RLAD offers a targeted approach to improving Q-value accuracy.

**Weaknesses:**

* This work seems a direct combination of anomaly detection models and offline RL. Although the idea is new, the paper presents few insights why choosing anomaly detection models instead of other techniques such as uncertainty estimated by ensemble Q networks. Detailed analysis and comparison should be performed.
* The writing and organization should be significantly improved. For example, there are large fractions of blanks in several pages. All the figures are not high-resolution.
* Several recent SOTA baselines are missed such as SVR[1], STR[2], SPOT[3] and CPI [4]. Without comparing the new SOTA algorithms, 'achieves near state-of-the-art performance' could be considered over-claimed.

[1] Supported value regularization for offline reinforcement learning. NeurIPS 2023.

[2] Supported trust region optimization for offline reinforcement learning. ICML 2023.

[3] Supported policy optimization for offline reinforcement learning. NeurIPS 2022.

[4] Iteratively Refined Behavior Regularization for Offline Reinforcement Learning. NeurIPS 2024.

**Questions:**

* How to train Anomaly Detection model? What's the training loss? How to obtain OOD samples?
* What's the different between other baselines that re-weight different state-action pairs such as DICE-based method SA-CQL [1]


[1] State-Aware Proximal Pessimistic Algorithms for Offline Reinforcement Learning.

---

### Official Review · Reviewer_r55T · 2024-11-01

**Soundness:** 3
**Presentation:** 3
**Contribution:** 3
**Rating:** 5
**Confidence:** 3

**Summary:**

The paper presents a novel approach to address the overestimation problem in offline reinforcement learning (RL) by leveraging anomaly detection techniques. The proposed method, Reinforcement Learning with Anomaly Detection (RLAD), utilizes anomaly scores to adjust the weights during the training process, thereby guiding the RL agent to focus on in-distribution data. The authors claim that this approach not only mitigates overestimation of out-of-distribution (OOD) actions but also achieves near state-of-the-art performance on continuous D4RL tasks.

**Strengths:**

- The paper introduces a creative solution by integrating anomaly detection with offline RL, which is a fresh perspective in addressing the overestimation issue, to my knowledge. This integration is both simple and flexible, allowing for potential adaptability with various RL algorithms and anomaly detection models.
- The experimental results demonstrate the effectiveness of the proposed method, showing improved performance over standard SAC and other baseline methods in several D4RL benchmark tasks. The use of anomaly scores to guide the learning process is empirically validated, highlighting the potential of this approach.

**Weaknesses:**

- The paper's use of anomaly detection to identify out-of-distribution (OOD) actions may not always be accurate. Anomalies aren't necessarily equivalent to OOD data; some anomalous state-action pairs might be crucial for achieving optimal policies (e.g., in a Markov Decision Process where the target state is reachable with a low probability). The paper doesn't adequately address how the method distinguishes between harmful OOD actions and beneficial anomalies.
- The experimental evaluation, while promising, is limited in scope. Important datasets and environments (e.g. other tasks in D4RL) are not evaluated. Additionally, the paper does not compare its approach against some relevant baseline methods, such as QT[R1], in offline RL, which could provide a more comprehensive understanding of its relative performance.
- The paper lacks a thorough theoretical analysis of why and how anomaly detection effectively mitigates overestimation in offline RL. A deeper exploration of the underlying mechanisms could strengthen the paper's contributions and provide insights into potential limitations or areas for improvement.

R1: Hu, S., Fan, Z., Huang, C., Shen, L., Zhang, Y., Wang, Y., & Tao, D. (2024). Q-value regularized transformer for offline reinforcement learning. arXiv preprint arXiv:2405.17098.

**Questions:**

Please refer to the weakness section.

---

### Official Review · Reviewer_E8s3 · 2024-11-02

**Soundness:** 2
**Presentation:** 1
**Contribution:** 2
**Rating:** 3
**Confidence:** 4

**Summary:**

The key challenge in offline RL is to avoid out-of-distribution (OOD) data during policy improvement. This work introduces a novel weighting mechanism to regulate the influence of in-distribution and OOD samples. Specifically, this weight is derived from anomaly detection models and is designed to assign lower values to OOD state-action pairs while maintaining higher values for in-distribution samples. Additionally, the proposed weight is applied in both policy evaluation and policy improvement.

**Strengths:**

The introduction of anomaly detection models into the offline RL setting is well-motivated and holds promise, though it requires pre-training an additional model and resembles many existing offline RL approaches. The process begins by training an anomaly detection model on the dataset, similar to early offline RL methods like BCQ. The resulting anomaly scores for dataset actions (and potentially for OOD actions) are then used as weights during policy evaluation and improvement steps. This approach aligns with uncertainty-based offline RL algorithms, such as UWAC.

**Weaknesses:**

**Conceptual errors**. Page 5, the weighted Q-function training: I believe there is an error in the gradient formulation. The expression given is, $\text{weight}(s', a')(Q_\theta (s,a) - r - Q^*(s',a')\nabla_\theta Q_\theta(s,a))$, but, based on standard RL textbooks, it should instead be,$\text{weight}(s', a')(Q_\theta (s,a) - r - \gamma Q^*(s',a')\nabla_\theta Q_\theta(s,a))$where the discount factor is missing. (I suppose this because the author compute the target y with a discount factor). Besides, page 6, line 300 the policy evaluation step: the action should be the policy action rather than the dataset action. What’s more, The gradient with the learning rate should be added to the policy’s parameter for gradient ascent, specifically in the form: $\theta = \theta + \alpha_\theta \nabla_\theta L_\pi$.  Such mistakes weaken the credibility of the paper.

**The advantage of using an anomaly detection model remains unclear**. I reviewed the explanation in Section 5.1, but it is challenging to see the connection between the empirical evidence presented and the conclusion that the proposed method outperforms CQL. For instance, the paper claims that *CQL shows … more frequent overestimations*. However, in Figure 2, it is actually the proposed method (normal_ours) that exhibits more instances of high  $Q_{\text{difference}} = \hat{Q}(s, a) - Q^*(s, a)$ . This observation suggests that the proposed method may also suffer from overestimation issues, contrary to the claimed advantage.

**Questions:**

I am inclined to rate this paper as “reject, not good enough,” for the following reasons:

1. Limited contribution and novelty: While introducing an explicit anomaly detection model into offline RL is somewhat novel, it closely mirrors concepts from existing approaches like BCQ and UWAC. The potential advantages of this additional pre-trained module are not entirely clear, making it difficult to see the unique impact of this contribution.
2. Conceptual errors: Certain conceptual issues, particularly in the policy improvement and evaluation steps, raise concerns about the paper’s reliability. These errors are surprising to see in a submission to ICLR, and addressing them would be essential for clarity and accuracy.
3. Presentation quality: The paper’s presentation could be improved by reducing wordy descriptions and enhancing the quality of figures.

---

### Official Review · Reviewer_k6Hz · 2024-11-02

**Soundness:** 3
**Presentation:** 3
**Contribution:** 3
**Rating:** 6
**Confidence:** 3

**Summary:**

This paper proposes leveraging anomaly detection methods to address the over-estimation of Q-values on out-of-distribution (OOD) actions. They assign an anomaly score for each state-action pair (s, a) and perform a weighted policy update in offline policy improvement.

**Strengths:**

This paper propose a new paradigm for offline RL. The method is simple yet effective.

**Weaknesses:**

The performance heavily depends on the anomaly detection module, which shifts the most critical part outside of the RL algorithm. I am curious in general which approach is better: more integrated offline RL methods or the proposed method with more separated modules. I am also interested in understanding how the choice of anomaly detection algorithm could affect the overall performance of the proposed method.

some typos
line 072:"neural network networks"

**Questions:**

I understand that this works for simple datasets, such as MuJoCo. But what if the states and actions are both in the form of language? I am curious about the potential adaptations of your method for language-based states and actions. For example, how might existing anomaly detection techniques in natural language processing be leveraged, and how would you define and measure OOD in a language context? Additionally, could you speculate on any challenges or limitations you foresee in applying your method to more complex data types?

---

### Note · Authors · 2024-11-21

**Comment:**

Thanks for your comments. We decide to withdraw our submission.
We will develop our paper according to your comments.

**Withdrawal Confirmation:**

I have read and agree with the venue's withdrawal policy on behalf of myself and my co-authors.